# A Biorefinery Approach to Biodiesel Production from Castor Plants

Fabiola Sandoval-Salas [1],*, Carlos Méndez-Carreto [1], Graciela Ortega-Avila [1], Christell Barrales-Fernández [1], León Raúl Hernández-Ochoa [2] and Nestor Sanchez [3]

1 Laboratorio de Investigación, Tecnológico Nacional de México/ITS de Perote, Km. 2.5 Carretera Federal Perote–México Col. Centro, Perote 91270, CP, Mexico; doc-201@itsperote.edu.mx (C.M.-C.); doc-007@itsperote.edu.mx (G.O.-A.); asistenteacad_01@itsperote.edu.mx (C.B.-F.)

2 Laboratorio de Química III, Facultad de Ciencias Químicas, Universidad Autónoma de Chihuahua, Circuito Universitario, Campus Universitario #2, Chihuahua 31125, CP, Mexico; lhernandez@uach.mx

3 Group of Energy, Material, and Environment, Faculty of Engineering, Universidad de La Sabana, Campus Universitario Puente del Común, km 7 Autopista Norte Bogotá, Bogotá 53753, CP, Colombia; nestor.sanchez1@unisabana.edu.co

\* Correspondence: investiga.itspe@gmail.com; Tel.: +52-(282)-1035973

**Abstract:** The high consumption of fossil fuels has significant environmental implications. An alternative to reduce the use of fossil fuels and develop ecological and economic processes is the bio-refinery approach. In the present study, the authors present the production of biodiesel from castor plants through a biorefinery approach. The process includes sub-processes associated with the integral use of castor plants, such as biodiesel production, oil extraction, fertilizer, and solid biomass production. Economic analyses show that producing only biodiesel is not feasible, but economic indicators (NPV, IRR, and profitability index) show it is much more feasible to establish businesses for the valorization of products and subproducts of castor plants, such as biomass densification. The internal rate return for the second scenario (E2) was 568%, whereas, for the first scenario (E1), it was not possible to obtain a return on investment.

**Keywords:** biodiesel; biorefinery; *Ricinus communis*

## 1. Introduction

Population increases and the high consumption of fossil fuels have caused a depletion of global fuel reserves. The development of environmentally friendly and economical alternatives to energy production is a priority [1,2]. Biofuels have become an alternative to tackle environmental damage caused by the excessive use of fossil fuels. One case is biodiesel production from castor oil, with the potential for mass production due to a high oil content in seeds, ranging from 30 to 60% [2–5].

Castor bean plants (*Ricinus communis* L.) adapt easily to the climate in Mexico. The plants are able to grow in different sites, including irrigation channels, landfills, vacant lots, and other areas under similar conditions [5–9]. It has been found that growth conditions affect the phenotype, adaptability, and oil yield of castor beans [5]. In Mexico, oil yields vary from 700 kg/ha to 2900 kg/ha for cultivation systems of castor bean plants, maize, and hybrid seed crops [10–13].

Castor oil extraction can be performed with mechanical methods, solvents, and with combined methods. The first method includes high-performance industrial presses [4,9]. In the second method, solvents allow the recovery of between 90 and 99% of oil. However, the use of solvents may be toxic [2,14]. Otherwise, combined methods incorporate mechanical and solvent extraction technologies to increase the process efficiency. Nonetheless, the selection of the technologies will depend on the production cost [15,16]. Similarly, different methods are used to increase the transesterification yield whose main purpose is biodiesel production and simultaneous separation from by-products such as glycerol, alcohol, and water. Another goal is that such by-products might be used in other suboperations [5,17].

For instance, the residual meal from castor oil extraction may contain between 20 and 40% of starch and up to 4.3% reducing sugars [5,8]; these materials may be used as a substrate in fermentation processes for the production of compounds of interest for the industry, such as ethanol and enzymes [12,18]. Bioconversion of such starchy substrates into ethanol encompasses a two-stage process (saccharification and fermentation): in the former, starch is converted into glucose with the use of enzymes ($\alpha$-amylase and glucoamylase), whereas in the latter stage, fermentative yeasts such as *Saccharomyces cerevisiae* can be used to produce ethanol from reducing sugar [18]. Technologies for ethanol production from starchy substrates are continuously developed, procuring the reduction of investment and operation costs while increasing ethanol yield [18].

The selection of alternatives depends on economic and technical analysis to determine the feasibility of investment. The feasibility study allows investors to make better decisions on the channeling of their assets; that is, optimization of capital in regards to the amount and cost, use of the most adequate production technology, to invest at the right time, determine the best location for the investment, and the best individuals to manage the business, among other aspects to be considered. The feasibility study is also a planning and development instrument, necessary to source private or public funding [12,18].

The feasibility study is the document containing the background and studies on an intended investment. The minimum aspects covered relate to the organization, administration, marketplace, technology, economics, and finances; and the analysis horizon, referring to the period to make projections, is generally in years. As a general criterion, the analysis horizon is longer than the term of an intended credit, if applicable, but equal or shorter than the life cycle of the main assets for the intended investment.

For that reason, the objective of this study is to assess the technical-economic feasibility of the processes developed for the integral exploitation of castor bean plants to obtain biodiesel and byproducts, with a biorefinery approach.

## 2. Materials and Methods

### 2.1. Process Design

Based on the studies conducted on castor bean plants, the following is a proposal for the processes that may be developed from castor bean crops, which are divided into 7 blocks depending on the product to be obtained.

Block 1. Castor plants

Based on the sampling of the plants in the mountainous region in the state of Veracruz, an analysis of the climate of each of the sites was conducted, and soil samples were taken for chemical composition analysis. Castor bean plant sprouts with flowers and fruit were collected from each site. The pressed plants were used for botanical identification using codes provided by the Instituto de Ecología A.C. from Xalapa (Veracruz), Mexico.

Block 2. Castor seeds

Phenotypes found in the zone under study were analyzed. Each phenotype was analyzed for composition [19].

Block 3. Refined oil

To obtain oil, seed extraction was made with a mechanical press, and later by using hexane as a solvent, in addition to combining methods. It was neutralized with phosphoric acid and sodium hydroxide solutions, to separate fatty residues that may be of commercial interest to the rubber and soap industries. The rest of the components dissolved in the oil are removed by centrifugation, and finally, the oil is washed to refine it [19].

Block 4. Biodiesel production

At this stage, the conditions for transesterification were defined [20], and in order to increase biodiesel yields, the biofuel separation and purification conditions were also studied. Byproducts such as glycerin have commercial uses, so it was conserved to be used in a subprocess.

Block 5. Castor meal

The effect of thermal and biochemical treatment was assessed for castor meal [21]. The presence of ricin was also determined. The application of the meal as fertilizer on potato crops at different growth stages was also assessed [22].

Block 6. Bioethanol production

The process of ethanol production from starchy substrates requires chemical or enzymatic hydrolyzation of raw material [23]. From the starch obtained from water washing the degreased castor meal, a 10% grout was prepared, which in turn was hydrolyzed with 34,000 MWU of α-amylase (ENMEX; MWU, Modified Wohlgemuth Unit: the amount of enzyme that will dextrinize 1 mg of soluble starch to a definite size dextrin in 30 min) at 90 °C, during 120 min and pH 6. The dextrinized broth was hydrolyzed again with 40 DU of glucoamylase (ENMEX; DU: the amount of enzyme that catalyzes the production of 1 g of glucose per hour) at 60 °C, for 24 h and pH 4 [24]. The hydrolyzed solution was adjusted to pH 4.5, then ammonium phosphate (0.3%) was added, and inoculated with 0.5 g of freeze-dried *Saccharomyces cerevisiae* yeast, and incubated at 30 °C, for 24 h. The obtained must was distilled and the alcohol content was quantified by densimetry with an alcoholometer.

Block 7. Solid biomass production

Residues from Blocks 1, 2, and 5 were analyzed for their potential use in the production of mushrooms and pellets (solid biomass).

Mass balance of the principal operations was conducted, and process diagrams were elaborated where the equipment required for each of the operations and unitary processes was included.

### 2.2. Financial Feasibility Assessment

For the financial assessment, a simulation was performed in Aspen Plus V11 (AspenTech®, Bedford, MA, USA) to determine the cost of the equipment and consequently the capital expenses (CAPEX) of the plant by using the lang factors. In addition, raw materials and product flow rates were obtained from Aspen to determine the operational expenses (OPEX) and the net present value (NPV) of the process. This software is employed to simulate bio-refineries concepts [25]. Overall process flow diagrams were developed, with operating conditions and mass-energy flows of the main systems. Simulation of the castor seed oil biorefinery was performed by Aspen Plus, as follows: Castor seed was modeled as a mixture of protein (20.78%), oil (50.20%), fiber crude (5.98%), ash (7.75%), and carbohydrates (7.96%). Protein was simulated as glutamic acid since it is the main amino acid present in the castor oil seed [26]. Fiber crude and carbohydrates were modeled as lignin and cellulose, respectively (Table 1). Properties of both lignin and cellulose were retrieved from Putsche and Wooley [27]. Ash was modeled as potassium since it is the main element present in castor seeds. Finally, castor oil was modeled as triricinolein and thermodynamic properties were obtained from Dimian et al. [28]. In addition, binary parameters for hexane and castor oil were retrieved by the UNIQUAC method using the regression tool in Aspen Plus. Data for liquid-liquid equilibria for these components were obtained from Baber et al. [29]. UNIQUAC parameters (i.e., r and q) were calculated through the Bondi method based on the chemical structure of castor oil. Besides, vapor pressure and latent heat vaporization parameters were obtained from the literature [30,31].

**Table 1.** Proximal composition of castor seed, castor beans, and castor husks.

| Component | Bean | Husk | Seed |
|---|---|---|---|
| Crude protein | $20.78 \pm 0.97$ | 13.37 | 19.69 |
| Oil | $50.20 \pm 0.47$ | 7.16 | 39.99 |
| Fiber crude | $5.98 \pm 0.20$ | 68.00 | 25.02 |
| Ash | $7.75 \pm 0.13$ | 11.47 | 9.30 |
| Carbohydrates | $7.96 \pm 0.00$ | 0.00 | 6.00 |

Table 2 shows the description of Aspen subroutines to produce biodiesel from castor seeds. Generally speaking, the simulation could be divided into five main sections, as shown in Figure 1. Firstly, the separation of castor seeds was simulated as SEP-1 to recover 69.83% of beans and 30.17% of husks. Specific recovery of components (e.g., protein, fiber, carbohydrates) were adjusted to determine the composition of seed beans as shown in Table 1. Afterward, castor beans were treated under two separation processes to recover the oil. On the one hand, a mechanical press whose oil recovery is 26%, and on the other, solvent extraction using hexane was used. The hexane flow rate was calculated using a design spec to give an overall oil recovery of 55%. The recovered oil was then converted into biodiesel at 60 °C according to Equation (1), where triricinolien ($C_{57}H_{104}O_9$) reacts with methanol ($CH_3OH$) to produce methyl ricinolein ($C_{19}H_{36}O_3$) and glycerol ($C_3H_8O_3$).

$$C_{57}H_{104}O_9 + 3CH_3OH \rightarrow C_{19}H_{36}O_3 + 3C_3H_8O_3 \tag{1}$$

**Table 2.** Aspen plus subroutines description.

| Equipment Name (Aspen Subroutine) | Description |
| --- | --- |
| Treshing (SEP-1) | Simulates the separation of castor bean and castor husk based on the fractional recovery of components. |
| Mechanic (SEP-1) | Simulates the separation of oil from castor beans based on the fractional recovery of 26% oil. |
| Mixer (Mixer) | Simulates the mixing process of castor bean with hexane. Hexane was calculated in order to achieve a global recovery of oil of 55%. |
| E-101 (Heater) | Adjust the temperature of the mix to 25 °C |
| Decanter (Decanter) | Simulates the separation of oil from the solid fraction and hexane. |
| E-102 (Heater) | Adjust the temperature to 100 °C to separate the hexane from the cake. |
| E-103 (Heater) | Adjust the temperature to 100 °C to separate the hexane from the oil. |
| Flash2 (Flash-2) | Separates adiabatically the oil and the hexane at vacuum pressure (0.2 bar). |
| P-101 (Pump) | Increases the pressure of the oil stream from 0.2 bar to 1.0 bar. |
| M-102 (Mixer) | Mixes the oil from the solvent-extraction and mechanical processes. |
| E-104 (Heater) | Adjust the temperature of the oil to the transesterification reaction temperature (60 °C). |
| M-103 (Mixer) | Mixes the ethanol and KOH to produce the methoxide required for the transesterification process. |
| E-105 (Heater) | Adjust the temperature of the methoxide to 60 °C. |
| Reactor (RStoic) | Simulates the transesterification process at 60 °C with complete conversion of oil into biodiesel. |
| E-106 (Heater) | Cools down the biodiesel from 60 °C to 25 °C. |
| M-104 (Mixer) | Mixes the glycerol with the biodiesel to ease the separation process. |
| Decant-2 (Decanter) | Simulates the phase separation of biodiesel and glycerol. |
| E-107 (Heater) | Adjust the temperature to 64 °C. |
| Flash-2 (Flash-2) | Separates adiabatically the methanol from the biodiesel under vacuum conditions (0.1 bar). |
| E-108 (Heater) | Cools down the biodiesel to 25 °C. |
| M1, M2, and M3 (Mixer) | Simulates the mixing stage during a washing process to remove biodiesel impurities. |
| Set-1, Set-2, and Set-3 (Decanter) | Simulates the phase separation process during the washing process to remove biodiesel impurities. |
| E-109 (Heater) | Adjust the temperature to 100 °C. |
| Flash-3 (Flash-2) | Separates adiabatically the water from biodiesel under vacuum conditions (0.1 bar). |
| Autoclave (Heater) | Simulates the detoxification process in an autoclave (121 °C) vapor fraction (0) |

Followed by the transesterification process, a purification step was done to remove all biodiesel impurities such as glycerol, methanol, KOH, and water. Initially, glycerol was added to enhance the phase separation between glycerol and biodiesel. Afterward, a vacuum distillation process was performed to remove methanol from the biodiesel mixture. Then, three extraction stages were added to remove KOH with water. Finally, water was separated from biodiesel by using a vacuum distillation process. Parallel to this process, the cake, yielded from the solvent recovery stage, was detoxified in an autoclave. Previously to this stage, hexane was removed. A detailed description of the simulation is shown in Table 2.

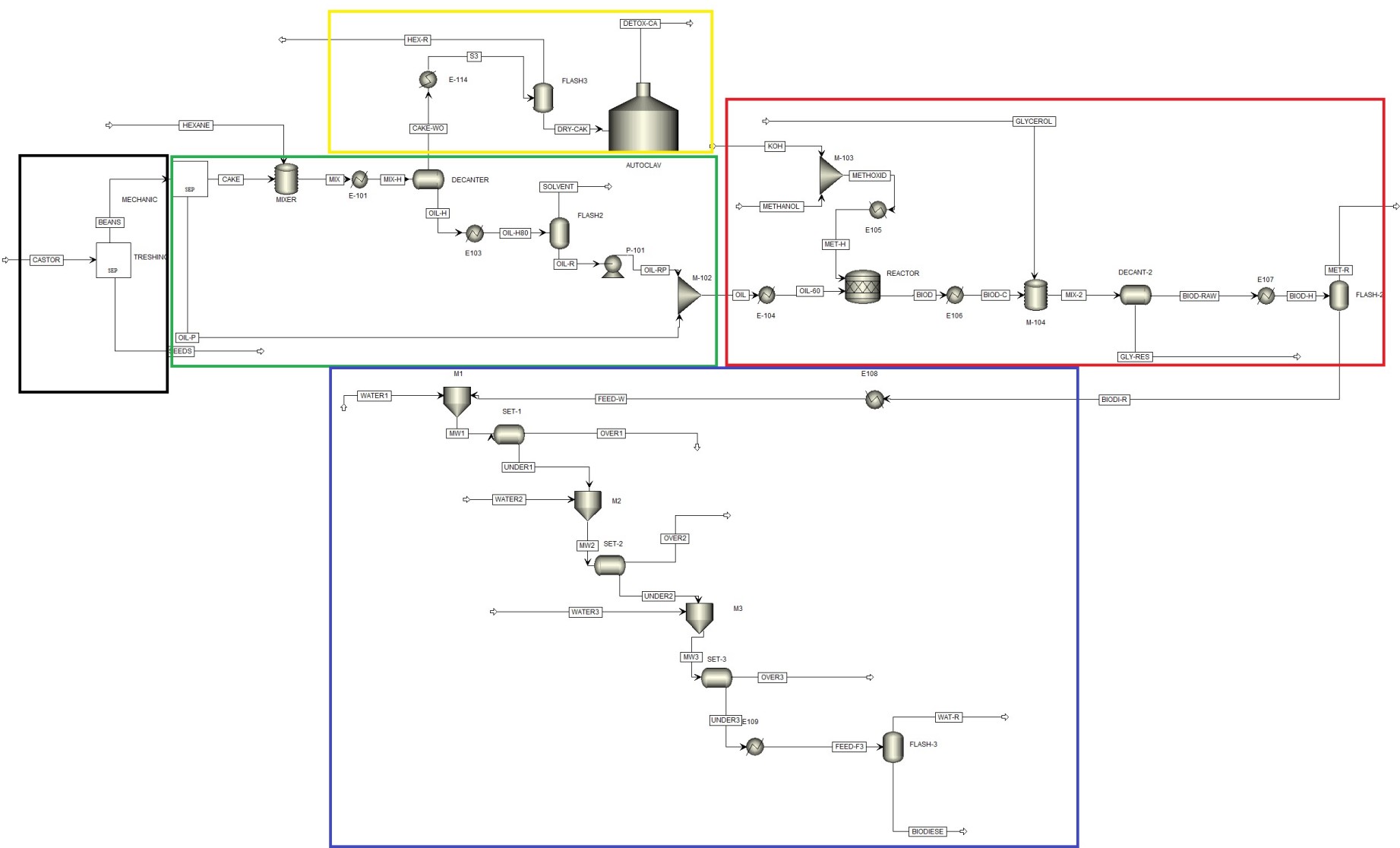

**Figure 1.** Aspen Plus simulation flowsheet. Castor seed separation (Black), oil separation (Green), biodiesel production (Red), biodiesel purification (Blue), and detoxification (Yellow).

### 2.3. Economic Analysis

A class-V economic analysis was performed to determine the economic feasibility of a castor seed biorefinery. In this economic analysis, two alternatives were considered. Scenario 1 only includes biodiesel production and Scenario 2 considers pellets, fertilizer, and biodiesel production. This kind of economic assessment is done to screen projects with an accuracy between 30% and 50% [32]. Therefore, CAPEX, OPEX, and NPV were considered. CAPEX and OPEX were calculated based on the equipment purchased and raw materials cost, respectively. In addition, lang factors were considered to calculate each of the items described in Table 3. Equipment purchased was calculated in Aspen Plus using the economic analyzer tools. All the equipment was sized and mapped. The raw materials cost was calculated according to Mexican suppliers.

**Table 3.** Estimate of diesel and biodiesel consumption.

| Diesel (%) | Biodiesel (%) | Monthly Biodiesel Consumption in Liters | Daily Biodiesel Consumption in Liters |
|---|---|---|---|
| 80 | 20 | 1,107,423.33 | 36,914.11 |
| 85 | 15 | 830,567.50 | 27,685.58 |
| 90 | 10 | 553,711.67 | 18,457.05 |
| 95 | 5 | 276,855.83 | 9228.53 |

## 3. Results

### 3.1. Process Design

The processes for elaboration of six products were proposed: refined castor oil, biodiesel, bioethanol, biogas, solid biomass, and edible mushrooms (Figure 2).

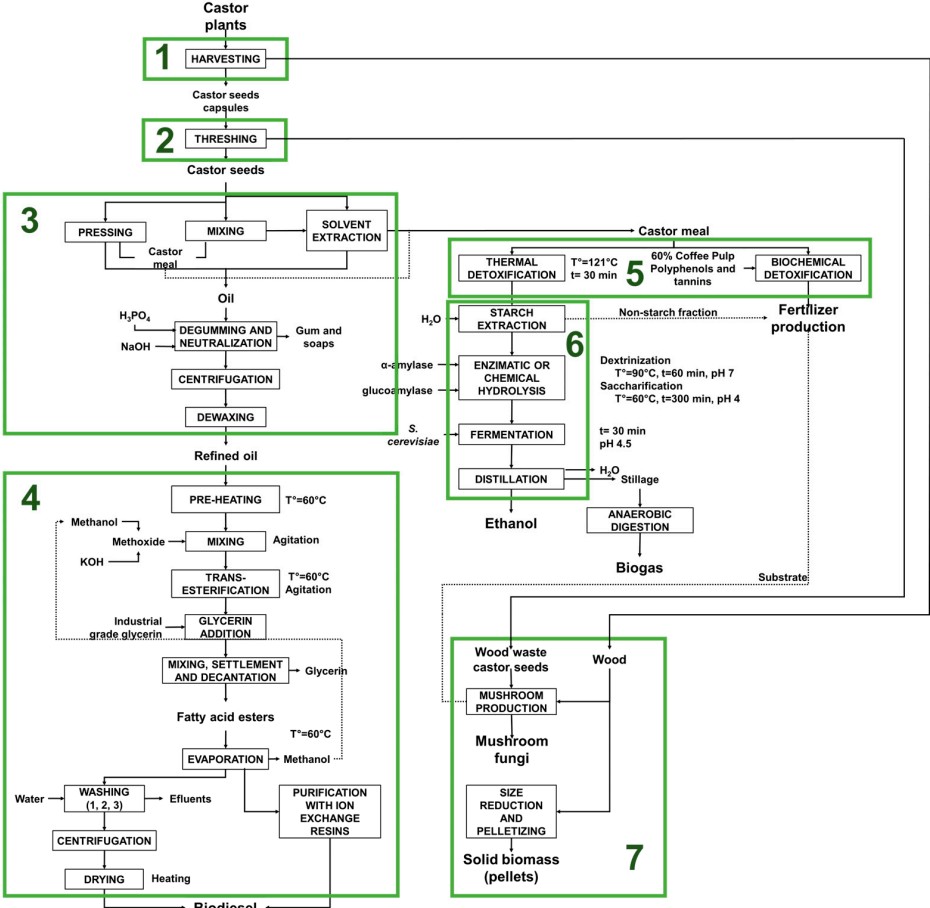

**Figure 2.** Proposal of exploitation of the species *Ricinus communis* for biodiesel and co-products.

Block 1. Castor plants

Significant differences were found between the soils at each site, and the temperature ranges from 12 to 22 °C, with high humidity. Soils had a high organic matter content, like beans, allowing the proper root development, making them adequate for agriculture, and viable for establishing castor bean crops [19].

Block 2. Castor seeds

The physical characterization of the castor bean seed phenotypes mentioned that they were long, oval seeds, averaging 10mm to 17mm, with black spots over a deep brown color, and there was no significant variation in the proximal composition of the phenotypes studied [19]. In general, for every 32.97 tons of capsules, up to 483.5 tons of stems and residues were obtained. After threshing, yields of 69.83% of seed were obtained, the rest corresponds to the husk, which can be used in other processes, such as manure and mushroom production.

Block 3. Refined oil

Through mechanical extraction, it was possible to recover up to 26% of oil contained in seeds, however, the oil quality was low due to some residual particles in the seeds; on the other hand, Soxhlet extraction using hexane as solvent provided better results of up to 52% of the total seed oil, obtaining a viscous, good quality oil, with intense yellow color, without apparent particles [20]. For the present study, the use of the combined process extraction was recommended.

Block 4. Biodiesel production

For biodiesel production at an industrial scale, the proposed process is as follows [19]:

Raw material: 60 L raw oil obtained by mechanical, solvent, or combined extraction, 40 L industrial-grade methanol, and 1.5 kg industrial-grade potassium hydroxide.

Preheating: To accelerate the transesterification process, oil is preheated to 60 °C; this operation decreases reaction time and also allows for decreasing oil viscosity.

Mixing: Hot oil is put in a tank fitted with agitation and/or recirculation; then, it is mixed with the (1.5 kg) potassium hydroxide and 40 L industrial-grade methanol.

Transesterification: In the transesterification process, oil triglycerides react with methanol in the presence of an alkaline catalyst, in this case, potassium hydroxide, producing methyl esters (such as ricinoleic acid ester, oleic ester, and linoleic acid ester) derived from fatty acids, and glycerin. The reaction is performed at 60 °C, using mechanical agitation at 100 rpm or recirculation for 120 min. During the process, it can be observed that the mixture becomes darker and viscosity decreases.

Glycerin addition, mixture, settlement, and decantation: With its high acid content, castor oil and its derivates have unique characteristics, such as polarity, high viscosity, specific gravity, and high solubility in alcohol. For processing 10,000 L of castor oil, 12 kg glycerin was added.

Evaporation: After the separation of the biodiesel phase, it is subjected to evaporation at 64 °C (preferably under vacuum conditions) to eliminate methanol residues.

Water washing: Biodiesel is subjected to triple water washing (the process works better if hot water between 60–90 °C is used) with the purpose of removing impurities and KOH residues, to conduct the washing approx. Then, 25 L of water is added, the mixture is agitated and centrifugated at 5000 rpm for 5 min, later the organic phase is separated, and the operation is repeated two times.

Drying: To eliminate water residues, biodiesel is heated to 100 °C, until the biodiesel stops bubbling and acquires a shiny hue.

Purification in ion exchange resin: As an alternative biodiesel purification process (in lieu of water-washing), an ion exchange resin can be used [21].

Block 5. Castor meal

For the use of residual castor meal, as the first step, it was necessary to evaluate their conditions and characteristics [33]. The component of interest in castor meal is ricin. This toxin could affect the use of castor meal in the subsequent processes. To eliminate ricin from castor meal, it was evaluated by thermal and biochemical treatments. Castor meal

thermal treatment under high-pressure (15 lb.) and for different times was evaluated, and it was found that as castor meal thermal treatment time increases, the intensity of the bands corresponding to molecular weight decreases. This is because heat breaks hydrogen bonds and hydrophobic protein interactions [34].

Detoxification for biochemical treatment was evaluated by free fermentation (the "Ogiris" method [35]), where it was observed that there is a decrease in intensity of electrophoresis bands corresponding to ricin protein after 6 h of cooking due to thermal treatment. This is indicative of ricin neutralization.

Block 6. Bioethanol production

Previous studies [34] indicate that it is required to separate the starch from the castor meal and hydrolyze it before formulating the culture medium for fermentation, as the meal alone may contain toxic compounds, including ricin, which during starch extraction are dissolved in water and used in washing, or they remain in the non-starchy section. Regarding ethanol production, yields are equal to 80% with *P. stipites* in hydrolyzed acid solution (1:6 ratio).

Block 7. Solid biomass production

Wood waste obtained from crushing seeds and reaping plants has the potential for exploitation in edible mushroom production and can also be used in pellet production with minimum investment. After evaluation, nitrogen content was shown to be slightly higher than most organic manures, which would be an advantage for using the meal as organic manure and mushroom production [36].

For each stage, the process operation conditions were defined, and considering the results obtained from the process experimental stage, the balances of matter of the main raw materials and products were performed (Figure 3).

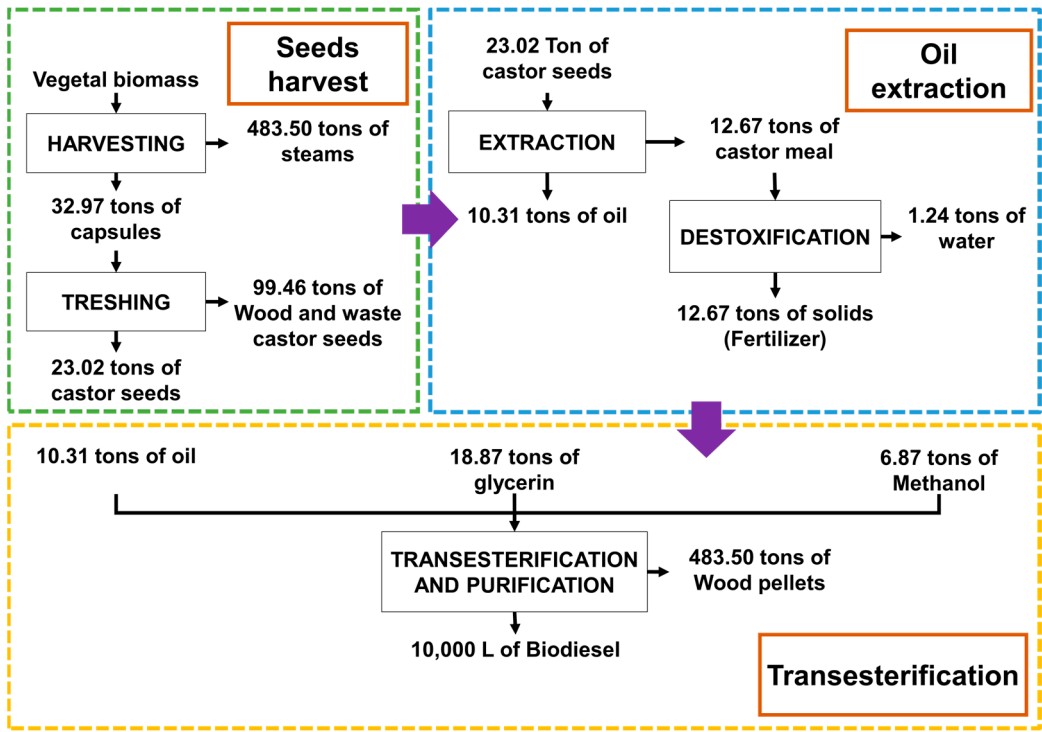

**Figure 3.** Balance of materials for castor bean co-product exploitation.

A proposal for the process equipment (Figure 4) and its layout diagram (Figure 5) were elaborated.

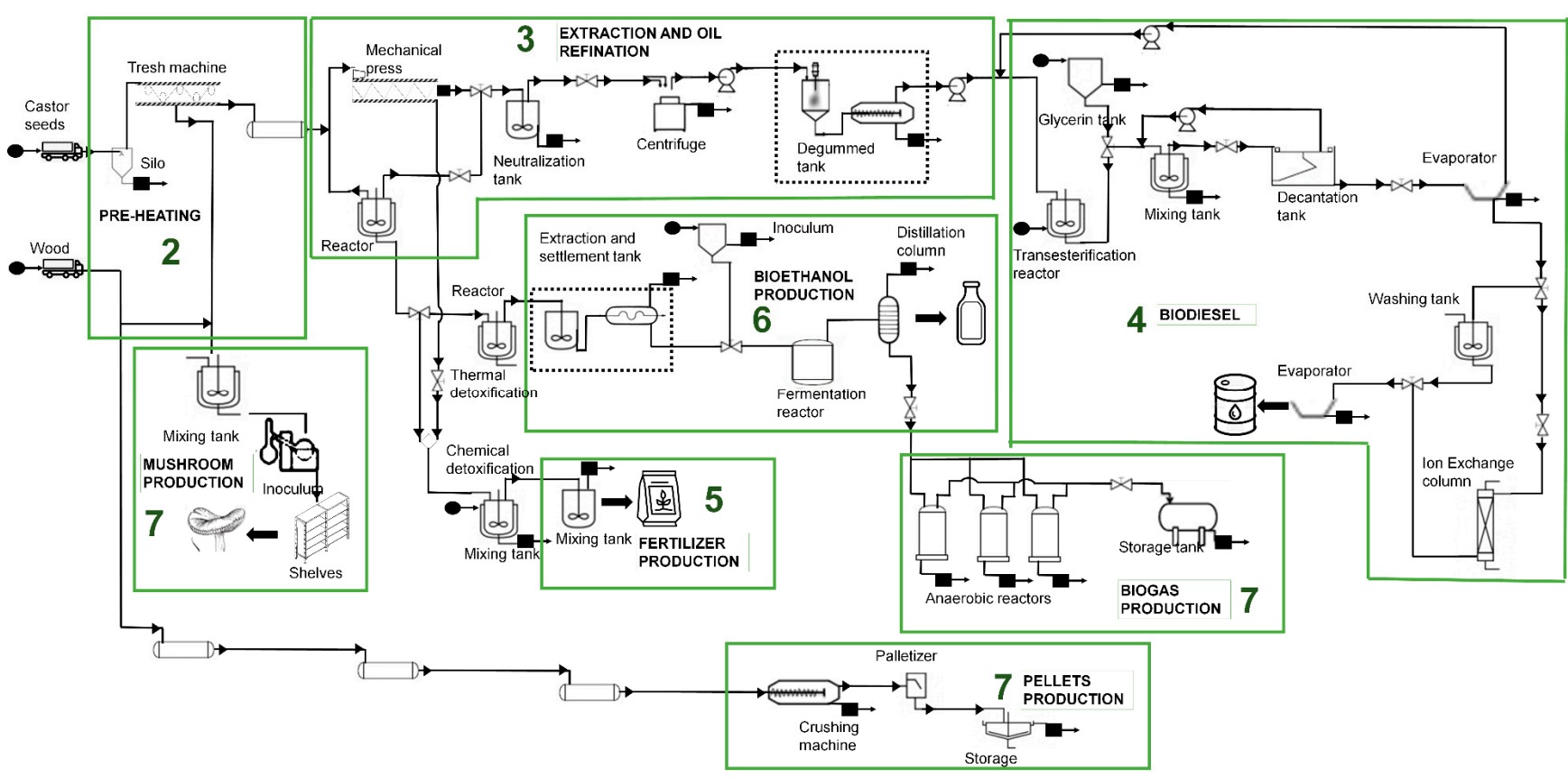

**Figure 4.** Diagram of equipment required for biodiesel and coproduct processing.

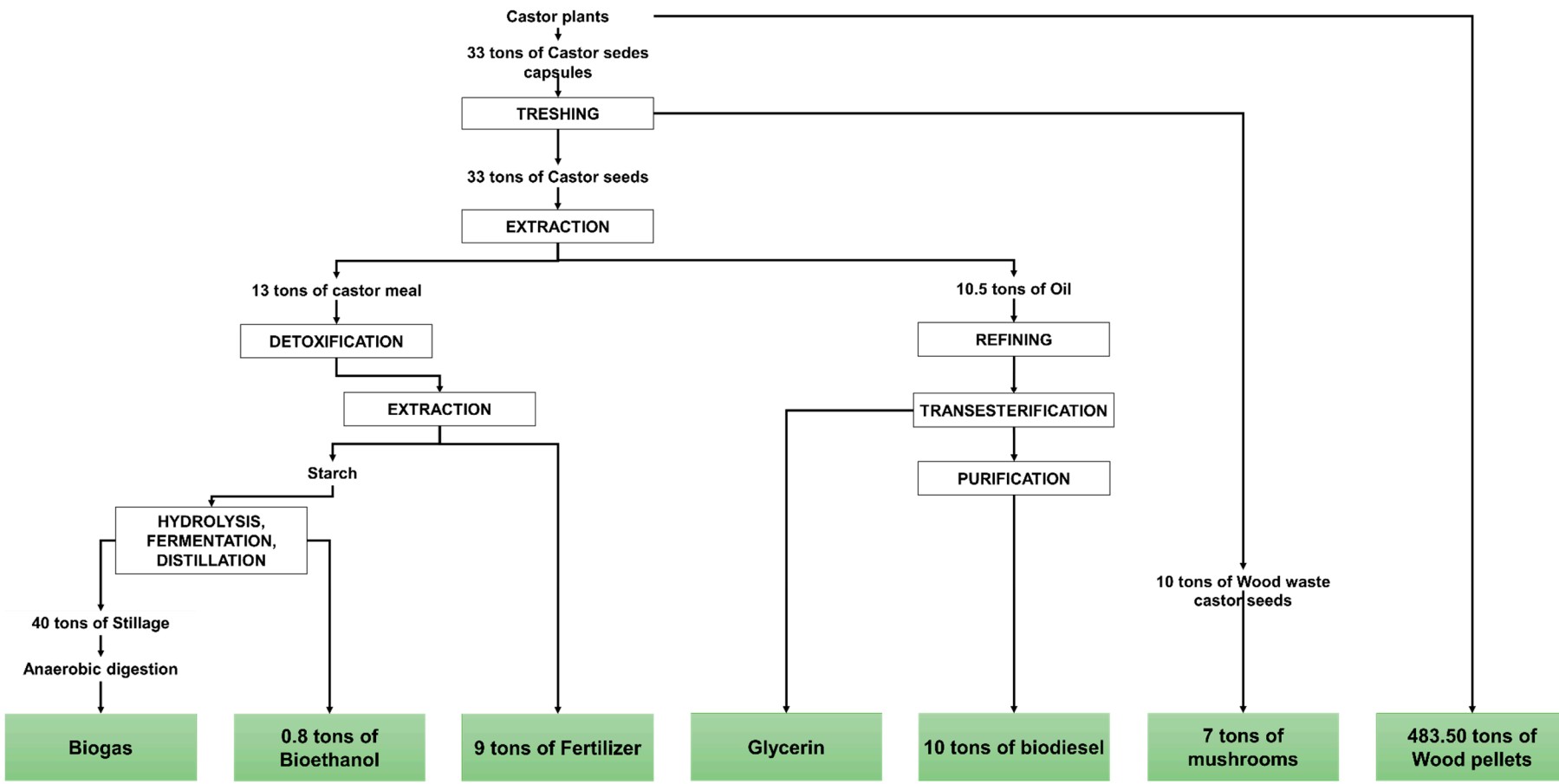

**Figure 5.** General balance of matter of processes developed in the project.

### 3.2. Technical-Economic Feasibility Study

The technical-economic determination of processes was estimated at a total of 55,3117 L of biodiesel per month. As biodiesel proportions in the final marketable fuel are 5 to 20% (Table 3), a 10,000 L per day facility (representing the minimum projected consumption) was considered.

Table 4 shows the inputs and outputs of main streams within a biorefinery scheme for castor seed conversion into biodiesel, pellets, and fertilizers.

**Table 4.** Input and outputs flow rates from the biorefinery of castor seeds.

| Input/Output | Value (kg/h) | Cost (USD/kg) |
|---|---|---|
| Castor seeds | 925.3 | 0.40 |
| Hexane | 331.4 | 1.40 |
| Methanol | 99.5 | 1.98 |
| KOH | 2.96 | 2.70 |
| Glycerol | 14.1 | 0.85 |
| Water | 237.9 | 0.0018 |
| Biodiesel | 137.5 (99.3 wt. %) | 4.0 |
| Castor husk | 279.2 | 1.0 |
| Fertilizer | 436.1 | 0.28 |

Table 5 shows the capital expenses for Scenario 1 and Scenario 2. For Scenario 1, the CAPEX was 3.3 MUSD whereas, for Scenario 2, CAPEX was 3.6 MUSD. Table 6 shows the operational expenses for both scenarios. In this case, OPEX was calculated in terms of raw materials consumption and this value was similar for both scenarios. Raw materials for biodiesel production represent the operational cost [37]. Hence, OPEX for both cases was 24 MUSD.

**Table 5.** Capital expenses for Scenario 1 and Scenario 2.

| Item | Scenario 1 | Scenario 2 | Lag Factor |
|---|---|---|---|
| Purchased equipment | $652,700.00 | $699,900.00 | 1.00 |
| Delivery | $65,270.00 | $69,990.00 | 0.10 |
| Purchased equipment installation | $254,553.00 | $272,961.00 | 0.39 |
| Instrumentation and controls | $147,510.00 | $158,177.00 | 0.23 |
| Piping | $202,337.00 | $216,969.00 | 0.31 |
| Electrical | $65,270.00 | $69,990.00 | 0.10 |
| Buildings | $189,283.00 | $202,971.00 | 0.29 |
| Yard Improvement | $78,324.00 | $83,988.00 | 0.12 |
| Service facilities | $358,985.00 | $384,945.00 | 0.55 |
| | $- | | |
| Engineering and supervision | $208,864.00 | $223,968.00 | 0.32 |
| Construction expenses | $221,918.00 | $237,966.00 | 0.34 |
| Legal expenses | $26,108.00 | $27,996.00 | 0.04 |
| Contractor's fee | $124,013.00 | $132,981.00 | 0.19 |
| Contigency | $241,499.00 | $258,963.00 | 0.37 |
| Working capital | $489,525.00 | $524,925.00 | 0.75 |
| CAPEX | $3,326,159.00 | $3,566,690.00 | |

With the CAPEX and OPEX values obtained from the Aspen Simulation, it was possible to calculate the plant profits and NPV. The selling price of the main products is shown in Table 1.

The biodiesel sale price was obtained considering production costs based on similar processes [38]. This represents a 340% markup from the average cost of diesel in Mexico [39]. The sale price of pelletized biomass was obtained from data provided by electronic commerce suppliers. The sale price of the castor meal ton was obtained based on the price of other similar organic fertilizers.

**Table 6.** Operational expenses for Scenario 1 and Scenario 2.

| Item | Scenario 1 | Lag Factor |
|---|---|---|
| Raw materials | $8,175,571.00 | 0.80 |
| Operating Labor | $1,021,946.00 | 0.10 |
| Direct supervisory | $102,195.00 | 0.10 |
| Utilities | $1,021,946.00 | 0.10 |
| Maintenance and repairs | $356,669.00 | 0.10 |
| Operating Supplies | $178,335.00 | 0.05 |
| Laboratory charges | $204,389.00 | 0.20 |
| Patents and royalties | $613,168.00 | 0.06 |
| Depreciation | $356,669.00 | 0.10 |
| Local taxes | $142,668.00 | 0.04 |
| Insurance | $ 35,667.00 | 0.01 |
| Rent | $356,669.00 | 0.10 |
| Financing | $356,669.00 | 0.10 |
| Administrative cost | $4,087,786.00 | 0.05 |
| Distribution and markets | $1,635,114.00 | 0.02 |
| Research and development | $4,087,786.00 | 0.05 |
| OPEX | $24,266,166.00 | |

For determination of financial indicators, we considered 360 days of operation per year for 10 years, with 100% of investment costs incurred in year zero. Th edepreciation coefficient was 10%, with a 30% tax rate and 12% discount rate, including 7.36% inflation index [40], plus a 4.64% risk premium (Table 7).

**Table 7.** Summary of financial indicators.

| Indicator | Scenario 1 | Scenario 2 |
|---|---|---|
| Net Present Value (USD) | −145,007.53 | 129,571.08 |
| Internal Rate of Return | - | 568% |
| Profitability Index | −43.6 | 36.3 |

## 4. Discussion

For Block 1 and Block 2, the castor plants belong to the species *Ricinus communis* and they grow in open sites, mainly in scrambled areas and on construction waste deposits, favoring seed development. In the proximal evaluation, we found that the oil content of seeds was between 39% and 48%, these values are lower than the 48.9% average values reported in seed collections in the state of Chiapas [8,14,41]. These characteristics favor the cultivation of castor plants in the central region of the country. Furthermore, the differences between castor seeds oil and protein content do not significantly affect the process. Castor plants are endemic to tropical and sub-tropical weather, so they do not require special conditions for growing; they do not compete with food crops [19,33].

However, in Block 3, oil values were found within the average range reported for this species, which may vary from 30–50% [5] to 50–60% [5,8,14]. Another key component evaluated in the seeds was total protein, which had values over 15%, matching the result obtained by Goytia-Jiménez et al. [10] for seeds collected in the state of Oaxaca. The protein content is deemed important as extraction meals generated from castor bean seeds have the potential for exploitation as soil fertilizers and for transformation into solid biomass. On the other hand, analysis of castor oil composition showed it contains mainly ricinoleic acid (80–90%), followed by minor concentrations of linoleic acid (3–6%), oleic acid (2–4%), and saturated fatty acids (1–5%) [5,8,14,34]. The best oil extraction method was the combined process (mechanical with solvent extraction); with this method, it is possible to recover up to 55.1% of oil contained in seeds. In addition, the refined oil has optimal characteristics for use in biodiesel production [19,20].

Then, in Block 4, with the transesterification of castor oil, there is no separation of biodiesel-glycerin phases (as it occurs commonly with the most used oils), making purification difficult. To avoid these issues, it is required that between 5 and 25% industrial-grade glycerin is added to the mixture coming from transesterification, after agitation, it is left to settle for 30 min to allow the formation of the two phases, which are separated by decantation. With the proposal process, it is possible to obtain 10,000 L of biodiesel, enough quantity to supply a medium-sized population [42].

For Block 5, regarding the use of detoxified castor meal as fertilizer, it has been found that nitrogen is one of the most important and complex constituents of plants [33]; plants capture this nutrient organically and inorganically [33]. Therefore, the denaturalization of protein would not affect product quality. On the other hand, the biochemical treatment includes mixing with tannins and polyphenols, and it was found that coffee pulp relatively decreases ricin concentration in castor meal, making it viable for use as fertilizer. Using castor meal as organic manure requires the application of prior treatments to detoxify it and avoid risks for the population and livestock. It is also important to determine its effectiveness as a tuber growth and/or development booster. *Solanum tuberosum* was used as a model to evaluate the effectiveness of the application of meal originated from castor oil extraction. Fertilization is particularly important for potatoes, as it affects growth and potatoes production [22]. Plant development requires high N and K contributions, and castor meal has high nitrogen contents. On the other hand, in tuberization, K demand is high, therefore this nutrient must be sufficiently available to provoke high mobilization of sugars from leaves to tubers [33].

In Block 6, for bioethanol, production costs increase, and bioethanol yields are relatively low; however, it is included in the process as an alternative for using residual castor meal [33].

For Block 7, the solid biomass production, the use of castor meal as manure has been reported and it has beneficial effects, including their high macronutrient content, increasing soil pH, reducing nematode presence, and increasing organic matter content. In a previous study, an increase in the yields of potato cultivars of 4 to 8 tons per cultivated hectare was found using castor meal, without side effects associated with the presence of ricin or the extraction method [22]. On the other hand, incorporating the castor meal into the potato crop growth substrate promotes up to a 50% increase in plant height compared with the maximum height of the witness treatment [22]. Meal fertilization showed better results than the application of chemical fertilizer, probably due to the high organic matter content. It is known that a good level of organic matter in the soil allows good water and oxygen permeability and improves nutrient mobility and cation exchange capacity, favoring multiplication and balanced life of beneficial microorganisms, allowing better root development, and favoring their breathing and nutrient absorption [33]. Castillo-González et al. [22] report that applying castor meal directly on potato fields increase potatoes yields from 4 to 8 tons per cultivated hectare (115–190 g potatoes per plant, considering 38,000 plants per hectare). No ricin bioaccumulation problems in tubers were found after 80 days of application.

In relation to the technical-economic feasibility study, this is a preliminary proposal informed by the generated and secondary information on diesel consumption, which refers to the regional market demand. Petroleum diesel blends in different proportions are also considered. Estimating a 5% biodiesel consumption, the facility is able to produce 10,000 L per day. Regarding the machinery and equipment needs for castor oil production, it is estimated that washing, crushing, refining, classification and correction, quality control, aeration, packing, sterilization, cooling, labeling, packaging, and storage of the end product will be performed at productive process areas. From the conversion factors used, for every 1000 kg of castor bean seed are obtained:

- 450 L biodiesel
- 471 L oil
- 550 kg fertilizer

- 302 kg mushrooms
- 21 m$^3$ pellets

The Net Present Value (NPV) obtained for Scenario 1 (E1) is (-) USD 145,007.53. Therefore, this scenario is rejected, as it indicates the present value of expenses is higher than the present value of income. That is to say that investing in the project has a lower rate than the opportunity interest rate. The project's Internal Return Rate (IRR) is not feasible, since this value represents the return rate value where NPV is 0. In addition, the Profitability Index is −43.6, indicating it is not enough to recover the investment [43].

For Scenario 2 (E2), the NPV is USD 129,571.08, which is considered an acceptable value as it indicates that the value of income is higher than expenses. That is, the investment in the project has a rate higher than the opportunity interest rate. Such a result is obtained by considering the income generated from selling the densified biomass (pellets). On the other hand, the IRR is 568%, equivalent to the return of investment, considering the value of the investment in a period of time. Such value indicates high feasibility for the project. This information is confirmed by the value of the Profitability Index, which is 36.3 [43]. Previous studies showed an IRR of at least 60% for castor oil seeds biorefinery [37].

**5. Conclusions**

The use of castor bean plants to obtain refined castor oil, biodiesel, bioethanol, biogas, solid biomass, and edible mushrooms becomes a feasible alternative as it does not compete with food crops or land use; thus, an equipment proposal was made for the processes and their layout diagram. The technical-economic determination of the processes was established as a total of 55,3117 L of biodiesel per month. Taking biodiesel proportions in the marketable end fuel to be from 5 to 30%, a 10,000 L per day facility was proposed, representing the minimum projected consumption.

The financial analysis showed that biodiesel production as a sole product is not feasible, making the biorefinery approach processing necessary. Economic indicators (NPV, IRR, Profitability Index) for products and co-products support the installation of facilities where all products and sub-products of a species are valued. The analysis of Scenario 2 supports the biorefinery approach for the comprehensive exploitation of castor plant crops, resource valorization, and decreasing the carbon footprint during biofuel production.

**Author Contributions:** Writing—review and editing, F.S.-S., C.M.-C., G.O.-A., C.B.-F., L.R.H.-O., N.S. All authors have read and agreed to the published version of the manuscript.

**Funding:** This research received no external funding.

**Institutional Review Board Statement:** Not applicable.

**Informed Consent Statement:** Not applicable.

**Data Availability Statement:** Not applicable.

**Conflicts of Interest:** The authors declare no conflict of interest.

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
