# Peer review of "A Biorefinery Approach to Biodiesel Production from Castor Plants"

_processes, doi:10.3390/pr10061208_

Round 1

Reviewer 1 Report

This study is an interesting study for biorefinery. Ricinus is easy to grow and is also suitable for use as a raw material for biomass. The explanation in the introduction is clear, and I think the series of processes is rational. Nitrogen and potassium have also been shown to affect leaf growth and root growth, respectively. Potassium is also said to improve the quality of potato starch. I think it makes a lot of sense to use caster meals as fertilizer. The work should be published. However, I think the following suggestions should improve the manuscript.

It is difficult to understand how to obtain technical-economic feasibility results. For example, although the processing time for each block is different, is it possible to secure the production volume required for one day? I understand that in the end, the authors have balanced the matter. But please explain in detail how you got the results that ensure it.

As a whole, many sentences are long, especially in the introduction. I recommend that the authors make it easier to read.

Regarding lines 179-181, the information has to arrange by combining liquid-liquid separation on lines 249-253 on page 9. It may not be suitable for described in the results section.

I think it's a good idea to change the layout of “bioethanol production” and “fertilizer production” in Figure 3 so that it correlates with Block 5 and Block 6 in Figure 1.

Could Figure 3 include biochemical treatment including mixing with tannins and polyphenols?

Author Response

Dear Reviewer,

We hope finds you well, we are thankful for your comments on the manuscript: "A biorefinery approach to biodiesel production from castor plants". We send you the modifications accord to the comments.

Kind regards,

PhD. Fabiola Sandoval Salas

Corresponding author.

Reviewer 2 Report

Main question addressed by the research: The work addresses the a biorefinery approach to biodiesel production from castor plants. However the title is not correct, as it does not summarize the main aspect related to "technical-economic" analysis.
Originality and relevance of the topic: The topic is relevant to the field, but the research gap to be covered is not clearly stated. What is the main contribution of this paper? It should be stated at the end of the Introduction.
Added value of the paper:  The manuscript takes into account a too general technical-economic analysis, however the main purpose of it is not clearly stated.

Quality of figures: Figures are good.
Specific improvements for the paper to be considered:

  1. Abstract is too short and general. It should summarize the main findings and applications of the paper.
  2. There is a big issue with scientific discussion in this paper. Results are just one table without clear background of the calculations and much more results and analysis of them are needed in order to reach high standards for publication. The paper is just descriptive and context until page 8.
  3. The authors refer to only one Table for the results. How were these estimations calculated? Unclear.
  4. The conclusions are poor and they would need more elaboration so they clearly match the results.

Author Response

(The authors gave the same response as above.)

Reviewer 3 Report

The authors attempted to show a technical-economic feasibility study of biodiesel production from castor bean seeds, under a biorefinery approach.

I opined that the manuscript does not meet the scientific requirements for publication. The manuscript lacks clarity and novelty. It is hard to follow the simulation and experimental investigation. The following issues are recommended to be addressed to improve the manuscript for possible resubmission.

  1. The abstract should not contain any citations. Avoid using citations in the abstract.
  2. Most of the references are > 5 years. More than half, 25++ of the references are > 10 years. Authors need to add more recent references for the research paper
  3. Generally, there is a lack of cohesiveness between the Materials and Methods, Results, and Discussion.  
  4. The material and methods section lacks clarifications on the manipulated and responding parameters.
    1. For section 2.1, line 83 – 125 for each of the blocks excluding block 6 there were no mentioned of standard analytical methods used and a hundred percent of the content was just the authors stating their general claims. i.e., line 104 – 105 the effect of thermal and biochemical treatment was assessed. What are the specifications for the treatments? What are the parameters assessed?
    2. For section 2.2 line 127 – 128 the authors should elaborate on the parameters for the economic feasibility study. The authors should specify the amount use as a basis and justify their choices. In line 130 – 131 the least squares method was mentioned but not elaborated. This should be elaborated further.
    3. The authors did not mention any analytical tools or software used to do the analysis. Authors should clarify the standard methods and analytical tools that are used.
  5. The results contained mostly claims by the authors. There were little to no comparison with other studies and for most of the results mentioned, the methods in which these measurements are taken were not specified in the material and methods sections.
    1. For Block 1 line 141 – 142 the number of sites and method of measuring the temperature were not specified in the materials and method section hence the data provided in this block seems arbitrary since it lacks context.
    2. For block 2 line 149 – 150 what does the author means by there’s no significant variation in the proximal composition of the phenotypes studied?
    3. For block 7. line 215 – 216 what does the authors mean when the operation conditions were defined and what was the results obtained from the experimental stage? The authors need to define the operation conditions and the results obtained from the experimental stage.
    4. The figures shown, figure 1 – 4 were not given full context and elaborated further which defeats the purpose of the figures to be added to the research paper. The authors need to give further elaboration on the figures shown in the study
    5. For section 3.2 line 229 where did this estimation of 55,137 liters of biodiesel per month comes from? line 229 – 230, 5 to 20 %? no references were given for this, and this was not mentioned previously in the materials and methods section. The authors need to clarify these numbers.
  6. The discussion contained the bulk of the content that should be in the results section
    1. Line 235 – 247 the analysis presented in this paragraph (i.e total protein, ricinoleic acid content) were not specified in the material and method section. The rest of the paragraphs are similar in which they contained analysis that was not mentioned in the materials and methods section
    2. Line 276 – 284 the authors made a lot of claims without referencing any standards. What is the “regional market demands”? How does the technical-economic feasibility study done by the authors compare to other studies? Need more clarifications and references for this paragraph.
  7. The authors have systematically proposed the biorefinery approaches, however, qualitative analyses are not found, and the quality/properties of the produced biodiesel are unknown. Qualitative studies such as GC-MS, FTIR, TGA, density of the oil, etc., should be performed to verify the effectiveness of the proposed approach.
  8. The authors should justify the effectiveness of KOH as the catalyst in transesterification. This result should be compared against acidic catalyst such as zeolite.
  9. The authors should investigate the concentration of KOH over the quality and quantity of the biodiesel produced.
  10. The written conclusion is not concise and is not informative.

Author Response

(The authors gave the same response as above.)

Round 2

Reviewer 1 Report

I confirmed the correction of the manuscript. I think this will be approved.

Author Response

We are thankful for your comments. Best regards.

Reviewer 2 Report

Paper has improved but more discussion for Tables 2 and 3 is needed in order to reach high standards for publication.

How were the costs estimated? Methodology should have been explained.

Author Response

We are thankful 

Reviewer 3 Report

The authors have made significant improvements on the manuscript.

However, the fundamental issue has not been addressed clearly which is the process simulation modelling of the Methodology.  It is suggested that the authors to refer the following article with regards to the Process Design  and Financial Feasibility Assessment.

1) Yeoh, L, and Ng, K. S. Future Prospects of Spent Coffee Ground Valorisation Using a Biorefinery Approach. Resources, Conservation & Recycling 179 (2022) 106123. (DOI: https://doi.org/10.1016/j.resconrec.2021.106123 )

Author Response

We are thankful for your comments. Best regards
